# Association of Handgrip Strength and Nutritional Status in Non-Dialysis-Dependent Chronic Kidney Disease Patients: Results from the KNOW-CKD Study

**DOI:** 10.3390/nu16152442

**Published:** 2024-07-26

**Authors:** Minsang Kim, Yeong-Won Park, Dha Woon Im, Yujin Jeong, Hyo Jin Noh, Soo Jin Yang, Eunjeong Kang, Hyunjin Ryu, Jayoun Kim, Ja-Ryong Koo, Ki Ryang Na, Eun Young Seong, Kook-Hwan Oh

**Affiliations:** 1Department of Internal Medicine, Seoul National University Hospital, Seoul 03080, Republic of Korea; kminsang71@gmail.com (M.K.); pyw1358@naver.com (Y.-W.P.); fomalhaut23@naver.com (E.K.); hdutopia@naver.com (H.R.); 2Department of Internal Medicine, Uijeongbu Eulji Medical Center, Uijeongbu 11759, Republic of Korea; dhawoon@naver.com; 3Department of Biostatistics, Korea University College of Medicine, Seoul 08308, Republic of Korea; wjddbwls1020@korea.ac.kr; 4Department of Food and Nutrition, Seoul Women’s University, Seoul 01797, Republic of Korea; yoon130623@naver.com (H.J.N.); sjyang89@swu.ac.kr (S.J.Y.); 5Department of Transplantation Center, Seoul National University Hospital, Seoul 03080, Republic of Korea; 6Medical Research Collaborating Center, Seoul National University Hospital, Seoul 03080, Republic of Korea; nunadli@gmail.com; 7Department of Internal Medicine, Hallym University Dongtan Sacred Heart Hospital, Hwaseong 18450, Republic of Korea; jrkoo@hallym.ac.kr; 8Department of Internal Medicine, Chungnam National University Hospital, Daejeon 35015, Republic of Korea; drngr@cnu.ac.kr; 9Department of Internal Medicine & Biomedical Research Institute, Pusan National University Hospital, Busan 49241, Republic of Korea; sey-0220@hanmail.net; 10Kidney Research Institute, Seoul National University Medical Research Center, Seoul 03080, Republic of Korea

**Keywords:** chronic kidney disease, nutrition, handgrip strength, malnutrition–inflammation score

## Abstract

Handgrip strength (HGS) is suggested as an indirect assessment of nutritional status in chronic kidney disease (CKD) patients, but evidence is limited for non-dialysis-dependent CKD (NDD-CKD) patients. This cross-sectional study included 404 patients from the Phase II KoreaN Cohort Study for Outcome in Patients With CKD. HGS, measured twice in each hand, was the exposure, and malnutrition status was defined by a malnutrition–inflammation score (MIS) of 6 or higher. A logistic regression analysis adjusted for age, sex, diabetes mellitus (DM), hypertension, CKD stages, smoking, overhydration, education, and income status was used to assess malnutrition risk. The predictability of HGS for malnutrition was evaluated using the area under the curve (AUC). Patients with lower HGS were older, had a higher prevalence of DM, and lower estimated glomerular filtration rate. Higher HGS was significantly associated with lower malnutrition risk after adjustment (per 1 standard deviation increase, adjusted odds ratio, 0.47 [0.30–0.75]). Subgroup analyses showed no significant interaction between HGS and malnutrition risk across age, sex, DM, and CKD stage. HGS showed fair predictability for malnutrition in men (AUC 0.64 [0.46–0.83]) and women (AUC 0.71 [0.55–0.86]). In conclusion, HGS is a useful diagnostic indicator of malnutrition in NDD-CKD patients.

## 1. Introduction

Chronic kidney disease (CKD) has the potential to cause malnutrition through various pathophysiologic mechanisms including metabolic acidosis, uremic toxins accumulation, systemic inflammation, and hormonal imbalance [1,2]. This subsequently increases morbidity and mortality in CKD patients [3,4]. Therefore, proper nutritional assessment and prevention of malnutrition are increasingly recognized as crucial aspects of care for CKD patients. However, given that the majority of previous studies have focused on the issue of protein energy wasting (PEW), which typically occurs in CKD patients at an advanced stage or undergoing dialysis [5], there is a paucity of research concerning the assessment of nutritional status in non-dialysis-dependent CKD (NDD-CKD) patients.

Among various nutritional assessment tools, handgrip strength (HGS) is a simple and reliable method for evaluating muscle function and overall physical fitness. In previous studies, HGS has been linked to a number of different diseases [6]. A low HGS is often indicative of diminished muscle quality and reduced muscle mass, which can result in metabolic dysregulation and insulin resistance [7,8]. In this regard, a low HGS has been shown to be associated with higher risks of type 2 diabetes mellitus (DM) [9,10], hypertension [7], and metabolic syndrome [11]. Given that a low HGS is associated with these metabolic diseases, previous studies have also shown that a low HGS is associated with an increased risk of cardiovascular diseases [12,13] and stroke [14]. Moreover, HGS has been used to evaluate physical status and activities associated with the upper limb in specific populations, including elderly participants [15] and those with cardiac diseases [16].

Particularly, in CKD patients, a low HGS was associated with a higher risk of mortality [17,18] and renal outcomes [19] in previous studies. In light of the established link between HGS and adverse outcomes in patients with CKD, HGS has been employed in a range of clinical settings, including the assessment of nutritional status in this patient population. The 2020 ‘Kidney Disease Outcomes Quality Initiative (KDOQI) Clinical Practice Guideline for Nutrition in CKD’, the most recent guideline for nutrition in CKD, suggests that HGS may be used as an indicator of protein-energy status and functional status in adults with CKD 1-5D [20]. However, the majority of the studies supporting this statement were conducted in patients receiving dialysis [21,22,23]. Consequently, there have been few studies investigating the association between HGS and nutritional status, specifically in NDD-CKD patients. 

Therefore, in this study, we aimed to investigate the association between HGS and nutritional status assessed by malnutrition-inflammation score (MIS) in the largest cohort of NDD-CKD patients in Korea. We hypothesized that NDD-CKD patients with a lower HGS would exhibit poorer nutritional status. 

## 2. Materials and Methods

### 2.1. Study Setting and Population

The KoreaN Cohort Study for Outcomes in Patients With Chronic Kidney Disease (KNOW-CKD), the largest multicenter prospective cohort study of NDD-CKD patients in Korea [24], has recently completed the enrollment for its Phase II study. The KNOW-CKD Phase II study (NCT03929900 at http://clinicaltrials.gov, first registered on 25 April 2019) is designed to strengthen the KNOW-CKD research as a resource on the health behaviors of CKD patients by collecting more information on their nutritional status and dietary patterns [25,26]. From 2019 through 2022, a total of 1490 patients aged 40–79 years, who were diagnosed with CKD stage 3–4 with an estimated glomerular filtration rate (eGFR) of 15–60 mL/min/1.73 m^2^, were enrolled. Patients with a diagnosis of glomerulonephritis or polycystic kidney disease, a previous history of dialysis, any organ transplant, heart failure (New York Heart Association functional classification III or IV), liver cirrhosis (Child-Pugh class B or C), or cancer, pregnant women, and those who had taken immunosuppressants within the past year were excluded from this study. Among the 1490 patients enrolled, 86 were excluded for the following reasons: 9 met the exclusion criteria, 3 were duplicate enrollments, 42 had insufficient baseline data, and 32 withdrew consent or refused to be followed within three months of registration. Thus, a total of 1404 patients were included in the baseline cross-sectional analysis of the KNOW-CKD Phase II study and longitudinal follow-up has been initiated. Among 1404 patients, 484 patients from four centers underwent a thorough nutritional assessment. Those with data missing for smoking status and nutritional assessment, including HGS, MIS, bioelectrical impedance analysis (BIA), and food frequency questionnaire (FFQ), were excluded. Furthermore, 3 patients with comorbidities that can significantly affect HGS were excluded, 1 with generalized myasthenia gravis, which is one of the autoimmune diseases; 1 with a history of carpal tunnel syndrome; and 1 with a history of quadriparesis. Finally, 404 patients were included in this cross-sectional analysis (Figure 1). 

### 2.2. Study Outcome and MIS

The primary outcome was malnutrition status, defined as MIS ≥ 6. MIS is a comprehensive and quantitative scoring system specific for the assessment of malnutrition and inflammation in CKD patients [27,28]. Furthermore, higher MIS was associated with an increased risk of mortality in NDD-CKD patients in previous studies [29,30]. The MIS has 10 components, comprising 7 components derived from the patient’s medical history and physical examination (weight change, dietary intake, gastrointestinal symptoms, functional capacity, comorbidity, fat store, and muscle wasting), body mass index (BMI), serum albumin, and total iron binding capacity. Each of the 10 components is graded on a scale of 0 (normal) to 3 (severely abnormal), and the total MIS, which ranges from 0 to 30, reflects the severity of malnutrition and inflammation. A higher MIS indicates a more severe degree of malnutrition and inflammation. Although a clear MIS criterion for defining malnutrition has not yet been established, we adopted the criterion of MIS ≥6, which has been commonly used in several previous studies [21,31,32]. Considering several prior researchers defined malnutrition as MIS ≥ 5 [33,34,35], the same analyses were performed with the criterion of MIS ≥ 5 as a sensitivity analysis.

### 2.3. Study Exposure and HGS

HGS was measured using a hand dynamometer (Takei 5401 Digital Dynamometer; Takei Scientific Instruments Co. Ltd., Niigata, Japan). Patients were instructed to sit in a chair and flex their elbow at a 90-degree angle. HGS was measured four times in total, twice on the right hand, followed by two measurements on the left hand. Each HGS measurement lasted approximately 3 to 5 s, with a rest period of about 15 s between measurements on the same hand. Subsequently, the maximum (HGSmax) and average (HGSavg) values of the four measurements were calculated. Since there is no standardized method to measure HGS in CKD patients, we primarily performed the analysis using HGSmax and additionally performed the same analysis using HGSavg as a sensitivity analysis. 

### 2.4. Data Collection

Demographic information including age, sex, smoking status, and comorbidities was collected by self-report and review of medical records. Anthropometric measurements including height, weight, waist circumference, and systolic and diastolic blood pressure were taken. BMI was calculated as weight (kg) divided by height squared (m^2^). Blood samples were obtained after overnight fasting to measure hemoglobin, albumin, C-reactive protein, glucose, and lipid profiles. Serum creatinine, urine creatinine, and urine microalbumin were measured at a central laboratory (Lab Genomics, Seoul, Republic of Korea). Serum creatinine levels were quantified by the isotope dilution mass spectroscopy–traceable method. The eGFR was calculated using the Chronic Kidney Disease Epidemiology Collaboration equation [36]. In nutritional assessment, additional anthropometric measurements including skinfold thickness (SFT) and mid-arm muscle circumference (MAMC) were obtained. The specific methodologies for these measurements are outlined in Method S1. Inbody S10 (Inbody Co., Ltd., Seoul, Republic of Korea), a multifrequency segmental BIA instrument, was used in this study to assess body fluid status including intracellular water (ICW), extracellular water (ECW), and total body water (TBW), and body composition parameters including body fat, fat-free mass (FFM), and soft lean mass (SLM). Overhydration status was defined as ‘ECW/TBW ratio > 0.39’ [37] and ‘TBW/FFM ratio > 0.74’ [38] using BIA data. Nutrient intake was assessed using a food frequency questionnaire and evaluated by nutrition experts.

### 2.5. Statistical Analysis

To evaluate the baseline characteristics, patients were divided into two groups based on their sex-specific median HGSmax value: low HGS and high HGS. The distributions of continuous variables were evaluated using the Shapiro–Wilk test. All continuous variables except systolic blood pressure were non-normally distributed and are presented as median (interquartile range) using Mann–Whitney U tests. Categorical variables are presented as frequencies (percentages) using χ^2^ test. Spearman’s rank correlation analysis was used to investigate the correlation between HGS and MIS. A multivariate correlation analysis was also conducted to evaluate the correlation between HGS and various nutritional parameters. Furthermore, linear and logistic regression analyses were conducted to investigate the association between HGS and MIS, as well as MIS ≥ 6 compared with MIS < 6 and MIS ≥ 5 compared with MIS < 5, respectively. Considering the potential confounding effects, a multivariable model, which was adjusted for age, sex, history of DM and hypertension, CKD stages, smoking status, overhydration status, and socioeconomic status including education and income status, was used in both linear and logistic regression analyses. Due to the missing data in income status, a sensitivity analysis was conducted following the imputation of missing values using the multivariate imputation by chained equations (MICE) method [39]. Additionally, subgroup analyses with stratification according to age (<60 or ≥60), sex, the presence of DM, and eGFR (<45 or ≥45 mL/min/1.73 m^2^) were conducted to compare the association of HGS and malnutrition in diverse subgroups. Finally, receiver operating characteristic (ROC) curve analysis was conducted to evaluate the predictability of HGS for malnutrition by calculating the area under the curve (AUC) values, and a comparison of the AUC was conducted using the Delong method. Optimal cut-off values of HGS to predict malnutrition in men and women were determined using the Euclidean index [40]. All statistical analyses were performed using R (version 4.3.2; The R Foundation for Statistical Computing, Vienna, Austria), and two-sided *p* values < 0.05 were considered statistically significant.

## 3. Results

### 3.1. Baseline Characteristics

The baseline characteristics of the 404 patients are presented in Table 1 according to the sex-specific median values of HGSmax. The median age of the total patients was 66 years, and the majority (72%) were men. The median eGFR of total patients was 39.5 mL/min/1.73 m^2^. The median HGSmax was 28.3 kg in the low HGS group and 37.5 kg in the high HGS group. Patients in the high HGS group were younger and had higher BMI, higher eGFR, lower albuminuria, lower prevalence of DM and cerebrovascular disease, and higher income and education status compared with the low HGS group. Additionally, patients with high HGS had higher hemoglobin and serum albumin levels and a lower fasting glucose level. In nutritional assessment, there was no significant difference in SFT or body fat measured by BIA between the low and high HGS groups. Nevertheless, patients with high HGS exhibited larger MAMC and higher FFM and SLM. Furthermore, patients with high HGS exhibited higher ICW, ECW, and TBW, although the ECW/TBW ratio and percentage of TBW/FFM were lower in patients with high HGS. There were no significant differences between the low and high HGS groups in total energy intake and macronutrients intake, including protein, fat, and carbohydrate. 

### 3.2. Correlation between HGS and Nutritional Parameters

A negative correlation (ρ = −0.14, *p* = 0.005) was observed between HGS and MIS (Figure 2). In multivariate correlation analysis, HGS was positively correlated with BMI, MAMC, ICW, ECW, TBW, FFM, SLM, and hemoglobin. Meanwhile, HGS was negatively correlated with biceps and triceps SFT (Appendix A).

### 3.3. Association between HGS and Malnutrition Status

Among a total of 404 patients, 23 patients (6%) were in a malnutrition status defined as MIS ≥ 6. Patients in the high HGS group exhibited a lower median MIS value and a lower proportion of MIS ≥ 6, compared with patients in the low HGS group. A higher HGS was significantly associated with a lower risk of malnutrition (per 1 standard deviation [SD] increase, odds ratio [OR], 0.47 [0.30–0.75]). In the multivariable model adjusted for age, sex, history of DM and hypertension, CKD stages, smoking status, overhydration status, education, and income status, a 1–SD increase in HGS was still associated with a decreased risk of malnutrition (Adjusted OR, 0.41 [0.19–0.89]) (Table 2). Additionally, when the MIS value was set as a continuous variable and the association between HGS and MIS was analyzed using linear regression, a higher HGS demonstrated a significant association with lower MIS in both the univariable and multivariable models.

### 3.4. Subgroup Analysis

In the subgroup analyses stratified by age, sex, presence of DM, and eGFR, the tendency of a negative association between HGS and the risk of malnutrition was consistently observed in all subgroups (Figure 3). Furthermore, the clinical factors of age, sex, DM, and eGFR did not show significant interactions with the association between HGS and malnutrition.

### 3.5. Sensitivity Analysis

In a sensitivity analysis, HGSavg was utilized as an additional study exposure and the criterion of MIS ≥ 5 was employed as an additional definition of malnutrition for the study outcome. A 1–SD increase in HGSavg was not significantly associated with a decreased risk of malnutrition, defined as MIS ≥ 6 or MIS ≥ 5 (Appendix A). Additionally, a 1–SD increase in HGSmax was not significantly associated with a decreased risk of malnutrition defined as MIS ≥ 5. However, among the 404 patients in the study population, 94 had missing data for their income status. Since the income status was included as a covariate in the multivariable adjustment model, the imputation was performed using the MICE method. In subsequent sensitivity analysis, both HGSmax and HGSavg showed significant negative associations with the risk of malnutrition, defined by MIS ≥ 6 or MIS ≥ 5, in both cases (Appendix A).

### 3.6. Predictability of HGS for Malnutrition

The AUC value of HGSmax as a predictor of malnutrition defined as MIS ≥ 6 was 0.64 (95% confidence interval [CI], 0.46–0.83) for men and 0.71 (95% CI, 0.55–0.86) for women (Figure 4). The optimized cut-off values of HGSmax for predicting malnutrition were 32 kg for men (sensitivity 62%, specificity 69%) and 20 kg for women (sensitivity 70%, specificity 69%). Additionally, there was no significant difference between the predictability of HGSmax and HGSavg for malnutrition in both men and women (Appendix A).

## 4. Discussion

In this cross-sectional study using the largest prospective cohort study of NDD-CKD patients in Korea, we demonstrated that lower HGS was significantly associated with a higher risk of malnutrition in NDD-CKD patients. Moreover, we demonstrated that this tendency was observed in all subgroups regardless of age, sex, DM, or CKD stage. Additionally, we demonstrated that HGS showed fair significance for the prediction of malnutrition both in men and women. Therefore, our study suggests that HGS is a useful diagnostic indicator of malnutrition status in NDD-CKD patients. 

To the best of our knowledge, our study represents the largest investigation of the association between HGS and malnutrition diagnosed by MIS among NDD-CKD patients. HGS has been widely used to assess muscle function in clinical studies due to its simplicity, ease of measurement, and cost-effectiveness [41,42,43]. Given that muscle function has been regarded as a useful predictor of malnutrition [44,45] and a more sensitive and applicable indicator of nutritional status alteration than anthropometric measurement [46,47], HGS has been widely used to evaluate nutrition status. Particularly in patients with CKD, various studies have been conducted regarding the association between HGS and nutritional status in patients receiving dialysis [21,22,23,48,49], since PEW, which is characterized by muscle mass wasting, is highly prevalent in kidney failure patients [50,51]. In the most recent guideline for nutrition in CKD patients, the 2020 KDOQI guideline, suggests at a grade 2B level that HGS may be used as an indicator of nutritional status even in NDD-CKD patients [20]. The cited study in this guideline for NDD-CKD patients identified a negative association between HGS and MIS using only correlation and linear regression analysis, indicating that a lower HGS is associated with worse nutritional status [28]. While another study additionally identified a negative correlation between HGS and MIS in NDD-CKD patients [52], there is still a lack of studies targeting NDD-CKD patients compared with those receiving dialysis. Given these circumstances, our study has the strength to provide additional evidence to support the recommendation in the 2020 KDOQI guideline using the largest cohort of NDD-CKD patients in Korea. 

Moreover, to the best of our knowledge, our study represents the first report on the predictability of HGS for malnutrition in NDD-CKD patients. Previous studies evaluated the predictability of HGS in identifying malnutrition, defined as MIS ≥ 6, among patients undergoing hemodialysis. Silva et al. [21] reported an AUC of 70% for men and 68% for women, while Sostisso et al. [53] reported an AUC of 73% for men and 61% for women. In comparison with the aforementioned studies, our findings indicate that HGS demonstrates fair predictability for assessing malnutrition, even in NDD-CKD patients, with AUC levels of 64% for men and 71% for women. Furthermore, our study proposes an optimal cut-off value for identifying malnutrition in NDD-CKD patients. Despite the extensive research on the association between HGS and malnutrition, HGS is not yet widely used in real-world clinical practice for diagnosing malnutrition. This is in contrast to its active use in diagnosing sarcopenia, where it is included in various diagnostic criteria with specified cut-off levels [54,55]. While several previous studies proposed cut-off values of HGS to diagnose malnutrition across various population groups [21,53,56,57,58,59], there have been no studies presenting cut-off values specifically tailored for NDD-CKD patients. Consequently, our study is meaningful as the first to propose cut-off values of HGS, 32 kg for men and 20 kg for women, for the assessment of malnutrition in this particular population. 

In a sensitivity analysis, we demonstrated that there was no significant difference in the predictability of HGSmax and HGSavg in assessing malnutrition. As HGSavg also demonstrated a significant negative association with the risk of malnutrition following the imputation of missing values for covariates related to income status, we proceeded to compare the predictability of HGSmax and HGSavg for the diagnosis of malnutrition status. In a recent systematic review of HGS measurements, these two methods were identified as the most frequent method to calculate HGS [60]. Although standardized methods for HGS measurement have not yet been established and our results did not show a significant difference in predictability between HGSmax and HGSavg, further research is warranted to compare HGS measurement methods and establish standardized protocols. The standardization of HGS measurement methods could potentially broaden their application in various fields. 

In the subgroup analysis stratified by the presence of DM, the negative association between HGS and the risk of malnutrition appeared to be attenuated in the group of patients with DM, although the presence of DM did not have a significant interaction with the association between HGS and malnutrition. Previous studies have shown that HGS tends to be lower in patients with DM [61,62]. In diabetic patients, peripheral neuropathy in both the upper and lower extremities is a common complication [62]. Furthermore, an abnormal cross-linking of collagen fibers, resulting from the accumulation of advanced glycosylation end-products, can lead to skin thickening and contracture in the hands of individuals with DM [63]. Thus, in patients with NDD-CKD and DM, a decline in HGS may occur regardless of nutritional status, potentially attenuating the association between HGS and malnutrition. Nevertheless, considering previous findings that have demonstrated a significant association between HGS and adverse outcomes such as mortality and cardiovascular events in patients with DM [64,65,66], further research is warranted to investigate the association between HGS and malnutrition in DM patients.

Our study has the strength of investigating the association between HGS and malnutrition status, defined by MIS, in the largest number of NDD-CKD patients to date. Furthermore, our study is the first to propose a cut-off value for HGS to predict malnutrition status in NDD-CKD patients. These strengths provide significant evidence for assessing nutritional status using HGS, especially in a context where methods for evaluating nutritional status in NDD-CKD patients are limited and existing evidence is scarce. This study has several limitations. First, due to the observational and cross-sectional nature of the study, a causal relationship between HGS and malnutrition cannot be established, and residual confounding factors and reverse causation may be present. Second, this study was conducted using data from a single country; therefore, further studies with diverse ethnic groups are necessary to expand the generalizability of the findings. Third, as mentioned in the methods section, a definitive MIS threshold for the diagnosis of malnutrition has not yet been established. Although we applied different MIS thresholds to assess malnutrition in our sensitivity analysis and obtained similar results, further research is necessary to investigate the criteria of MIS associated with adverse outcomes in NDD-CKD patients through longitudinal analysis.

## 5. Conclusions

In conclusion, HGS can serve as a valuable diagnostic indicator of malnutrition status in NDD-CKD patients. Clinicians could detect malnutrition earlier in patients with NDD-CKD by using HGS, a simple and reliable measurement.

## Figures and Tables

**Figure 1 nutrients-16-02442-f001:**
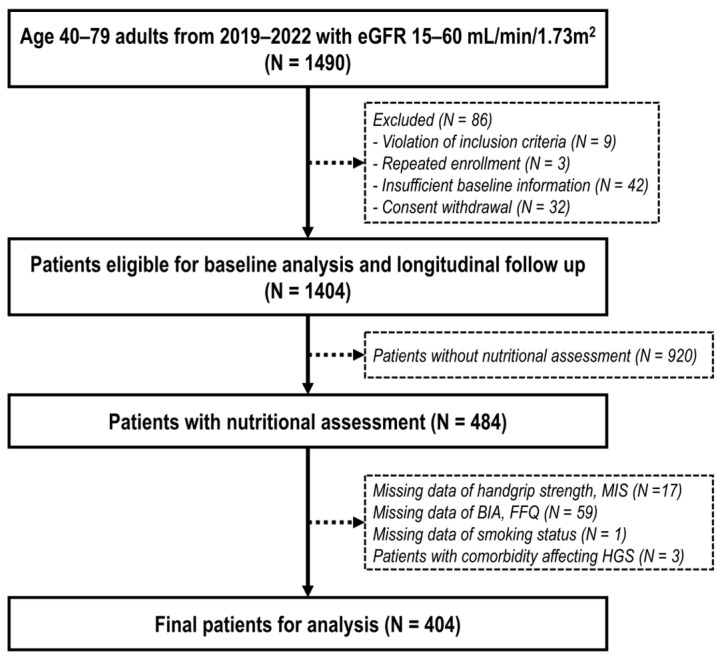
Flow chart of the study population.

**Figure 2 nutrients-16-02442-f002:**
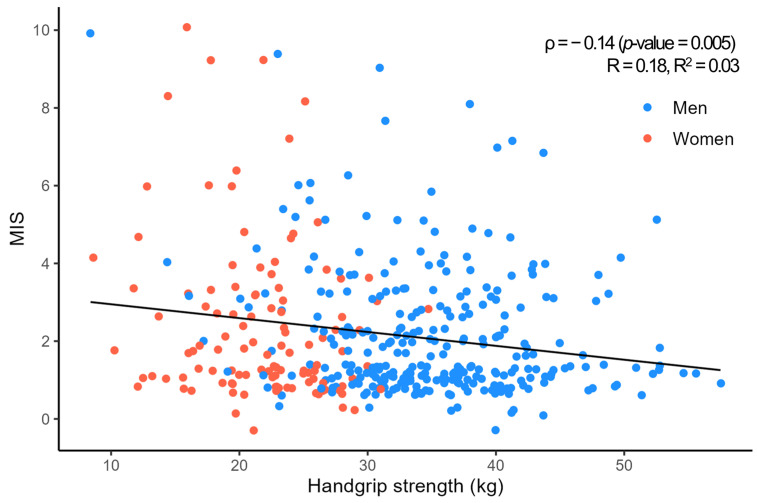
Correlation between handgrip strength and malnutrition–inflammation score. The x-axis indicates the handgrip strength (kilogram). The y-axis indicates the malnutrition–inflammation score. Correlation was analyzed by Spearman’s rank correlation analysis. The black line represents the trend line showing the direction and strength of the correlation derived from linear regression analysis. MIS, malnutrition–inflammation score.

**Figure 3 nutrients-16-02442-f003:**
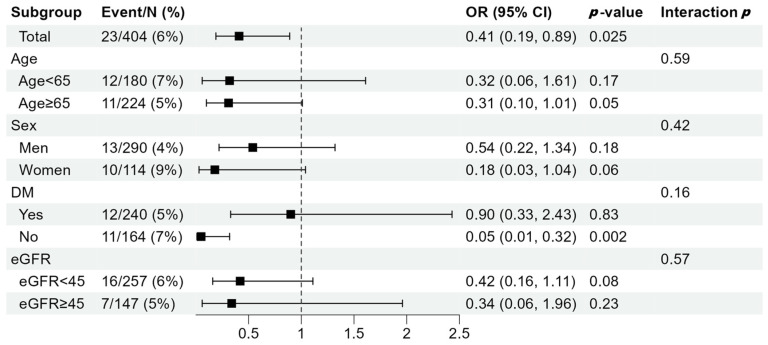
Subgroup analysis about the risk of malnutrition status according to handgrip strength. Subgroup analyses were conducted with stratification according to age (<60 or ≥60), sex (men or women), the presence of DM (yes or no), and eGFR (<45 or ≥45 mL/min/1.73 m^2^). The odds ratios were standardized with 1 standard deviation increase in handgrip strength, and were adjusted for age, sex, history of DM and hypertension, CKD stages, smoking status, overhydration status, education, and income status. The black squares represent the adjusted odds ratio. DM, diabetes mellitus; eGFR, estimated glomerular filtration rate; OR, odds ratio; CI, confidence interval.

**Figure 4 nutrients-16-02442-f004:**
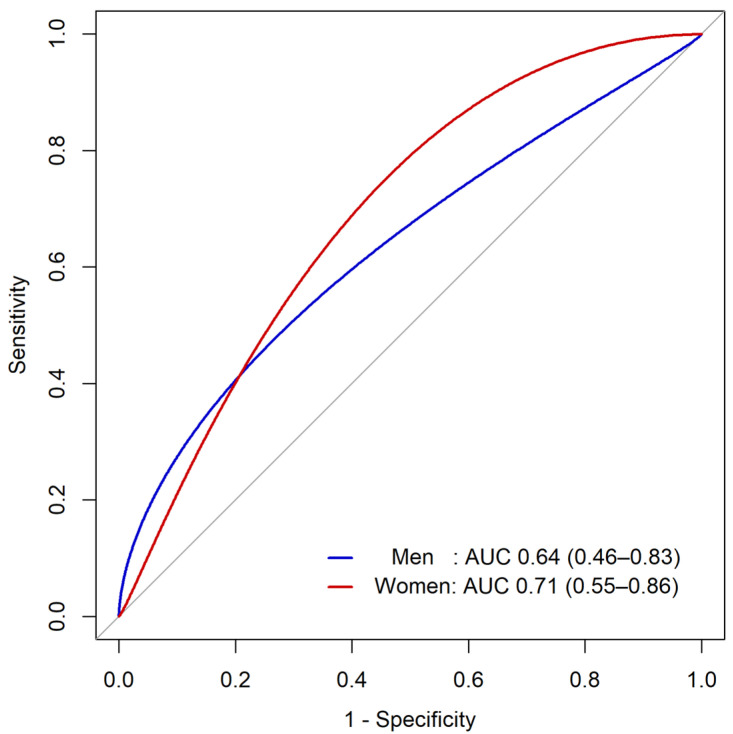
Predictability of handgrip strength for diagnosing malnutrition in men and women. Handgrip strength (HGS) is the maximum value of handgrip strength measured twice on each hand. Receiver operating characteristic (ROC) curves of HGS for assessing malnutrition status, defined as malnutrition–inflammation score ≥ 6, were plotted. Area under the curve values (AUC) were calculated. The x-axis indicates 1 – specificity, and the y-axis indicates sensitivity.

**Table 1 nutrients-16-02442-t001:** Baseline characteristics of the study population according to the sex-specific median value of handgrip strength.

	Total (N = 404)	Low HGS (N = 202)	High HGS (N = 202)	*p*-Value
	N (%)	N (%)	N (%)	
Age (years old)				<0.001
40–49	18 (4%)	1 (0.5%)	17 (8%)	
50–59	87 (22%)	28 (14%)	59 (29%)	
60–69	167 (41%)	82 (41%)	85 (42%)	
70–79	132 (33%)	91 (45%)	41 (20%)	
Sex				1.00
Men	290 (72%)	145 (72%)	145 (72%)	
Women	114 (28%)	57 (28%)	57 (28%)	
Monthly income (KRW)				<0.001
<1.5 million	61 (20%)	32 (22%)	29 (18%)	
1.5–4.5 million	168 (54%)	93 (63%)	75 (46%)	
>4.5 million	81 (26%)	22 (15%)	59 (36%)	
Education status				0.008
<High school	108 (27%)	66 (33%)	42 (21%)	
High school	136 (34%)	69 (34%)	67 (33%)	
>High school	160 (40%)	67 (33%)	93 (46%)	
Cigarette smoking status				0.61
Never smoker	158 (39%)	79 (39%)	79 (39%)	
Current smoker	44 (11%)	19 (9%)	25 (12%)	
Former smoker	202 (50%)	104 (51%)	98 (49%)	
Alcohol drinking status				0.53
Less than monthly	199 (60%)	105 (63%)	94 (57%)	
Less than weekly	117 (35%)	54 (33%)	63 (38%)	
More than weekly	14 (4%)	7 (4%)	7 (4%)	
Comorbidity				
Diabetes mellitus	240 (59%)	142 (70%)	98 (49%)	<0.001
Hypertension	363 (90%)	181 (90%)	182 (90%)	1.00
Coronary artery disease	56 (14%)	30 (15%)	26 (13%)	0.51
Cerebrovascular disease	52 (13%)	35 (17%)	17 (8%)	0.012
CKD stage				0.005
Stage 2	19 (5%)	3 (1%)	16 (8%)	
Stage 3a	128 (32%)	56 (28%)	72 (36%)	
Stage 3b	156 (39%)	86 (43%)	70 (35%)	
Stage 4	100 (25%)	56 (28%)	44 (22%)	
Stage 5	1 (0.2%)	1 (0.5%)	0 (0%)	
Overhydration ^1^	71 (18%)	47 (23%)	24 (12%)	0.004
MIS (Category) ^2^				
≥ 6	23 (6%)	16 (8%)	7 (3%)	0.09
≥ 5	39 (10%)	24 (12%)	15 (7%)	0.18
	**Median (IQR)**	**Median (IQR)**	**Median (IQR)**	
Age (years old)	66 [59, 71]	69 [63, 74]	62 [55, 68]	<0.001
BMI (kg/m^2^)	25.2 [23.2, 27.6]	24.8 [22.6, 26.9]	25.8 [23.7, 28.1]	0.001
Waist circumference (cm)	90.0 [83.7, 96.0]	89.0 [83.0, 96.0]	91.0 [84.0, 96.0]	0.19
Blood pressure (mmHg)				
Systolic ^3^	132 ± 16	133 ± 16	131 ± 16	0.26
Diastolic	74 [68, 83]	72 [66, 82]	75 [69, 84]	0.06
Nutritional assessment				
Handgrip strength (kg)				
HGS maximum	31.3 [25.0, 37.5]	28.3 [20.5, 31.6]	37.5 [29.4, 41.4]	<0.001
HGS average	28.8 [22.4, 34.3]	25.0 [18.6, 29.1]	34.3 [26.7, 39.0]	<0.001
Skinfold thickness (mm)				
Biceps	10.1 [7.6, 14.0]	10.2 [7.8, 14.1]	9.6 [7.6, 13.7]	0.34
Triceps	14.3 [11.2, 18.8]	14.2 [11.2, 17.1]	14.6 [11.2, 19.8]	0.24
Subscapular	18.8 [15.8, 22.0]	18.4 [15.8, 21.5]	18.9 [15.8, 22.3]	0.52
Suprailiac	18.9 [14.2, 24.1]	18.5 [14.0, 23.6]	19.5 [14.7, 24.2]	0.52
MAMC (cm)	23.5 [21.4, 25.3]	22.9 [21.0, 24.4]	24.4 [21.6, 26.1]	<0.001
BIA				
ICW (Liter)	23.1 [19.5, 25.9]	21.8 [18.9, 24.4]	24.4 [20.6, 27.7]	<0.001
ECW (Liter)	14.7 [12.4, 16.4]	13.9 [12.0, 15.6]	15.3 [13.1, 17.2]	<0.001
TBW (Liter)	37.7 [31.9, 42.5]	35.7 [30.8, 39.9]	39.7 [33.8, 45.1]	<0.001
ECW/TBW (ratio)	0.39 [0.38, 0.39]	0.39 [0.39, 0.40]	0.38 [0.38, 0.39]	<0.001
TBW/FFM (%)	73.8 [73.5, 74.0]	73.8 [73.6, 74.0]	73.7 [73.5, 74.0]	0.037
Body fat (kg)	17.9 [13.6, 22.9]	17.9 [13.1, 22.8]	17.9 [14.1, 22.9]	0.36
FFM (kg)	51.0 [43.2, 57.7]	48.5 [41.7, 54.0]	54.0 [45.9, 61.2]	<0.001
Soft lean mass (kg)	48.3 [40.8, 54.4]	45.7 [39.5, 51.0]	51.0 [43.5, 58.0]	<0.001
MIS (Continuous)	1 [1, 3]	2 [1, 3]	1 [1, 3]	0.046
Dietary intake				
Total energy (kcal/day)	1383 [1074, 1684]	1370 [1035, 1655]	1402 [1135, 1687]	0.31
Protein (g/day)	41.4 [31.8, 53.1]	39.9 [31.3, 52.8]	42.8 [33.0, 53.2]	0.34
Protein (% of E)	12.2 [11.1, 13.5]	12.3 [11.1, 13.6]	12.2 [11.1, 13.5]	0.64
Fat (g/day)	25.8 [16.9, 38.4]	25.3 [15.3, 37.0]	26.3 [18.1, 38.6]	0.36
Fat (% of E)	17.3 [13.0, 22.6]	17.8 [13.0, 22.6]	17.2 [12.8, 22.5]	0.86
Carbohydrate (g/day)	240.3 [178.0, 284.3]	239.2 [170.4, 284.0]	241.7 [186.9, 284.6]	0.39
Carbohydrate (% of E)	68.8 [62.6, 74.2]	68.3 [63.1, 74.0]	69.4 [61.9, 74.3]	0.96
Kidney function				
Creatinine (mg/dL)	1.6 [1.4, 2.1]	1.7 [1.4, 2.1]	1.6 [1.3, 2.0]	0.25
eGFR (mL/min/1.73 m^2^)	39.5 [30.1, 49.6]	38.0 [29.6, 46.7]	41.6 [31.6, 52.2]	0.011
UACR (mg/g)	210.2 [41.3, 855.7]	234.6 [58.3, 902.5]	182.4 [24.8, 844.1]	0.041
Laboratory measurements				
Hemoglobin (g/dL)	12.5 [11.6, 13.9]	12.2 [11.3, 13.5]	13.1 [11.7, 14.3]	<0.001
Albumin (g/dL)	4.3 [4.0, 4.4]	4.2 [4.0, 4.4]	4.3 [4.1, 4.5]	0.05
Total CO_2_ (mmol/L)	26.0 [23.2, 28.0]	25.0 [23.0, 27.2]	26.0 [24.0, 28.0]	0.09
CRP (mg/L)	0.6 [0.3, 1.5]	0.7 [0.3, 1.8]	0.6 [0.4, 1.3]	0.14
Fasting glucose (mg/dL)	110.0 [98.0, 130.5]	114.5 [99.0, 144.0]	106.0 [97.0, 123.0]	0.005
Total cholesterol (mg/dL)	152.0 [133.0, 180.0]	150.0 [132.0, 178.0]	153.0 [133.0, 185.0]	0.32
Triglyceride (mg/dL)	125.0 [95.0, 187.0]	125.0 [95.0, 187.0]	129.0 [95.0, 180.0]	0.62
HDL cholesterol (mg/dL)	44.0 [37.0, 52.0]	43.0 [36.0, 53.0]	45.0 [37.0, 52.0]	0.28
LDL cholesterol (mg/dL)	83.0 [66.0, 103.0]	79.0 [64.5, 102.0]	87.0 [66.0, 107.0]	0.10

Data are presented as the median (interquartile range) for continuous variables except systolic blood pressure or number (%) for categorical variables. HGS, handgrip strength; CKD, chronic kidney stage; MIS, malnutrition–inflammation score; BMI, body mass index; MAMC, mid-arm muscle circumference; BIA, bioelectrical impedance analysis; ICW, intracellular water; ECW, extracellular water; TBW, total body water; FFM, fat-free mass; E, energy; eGFR, estimated glomerular filtration rate; UACR, urine albumin-creatinine ratio; CRP, C-reactive protein; HDL, high-density lipoprotein; LDL, low-density lipoprotein. 1. Overhydration was defined as ECW/TBW > 0.39 and TBW/FFM > 74%. 2. MIS (≥5 vs <5; ≥6 vs <6) was demonstrated as the ‘Number (%)’. 3. Systolic blood pressure is presented as the mean (1 standard deviation).

**Table 2 nutrients-16-02442-t002:** Association between handgrip strength * and malnutrition–inflammation score.

	Unadjusted		Age-Sex Adjusted		Multivariable Model ^1^		Multivariable Model ^2^	
	**β (95% CI)**	** *p* **	**β (95% CI)**	** *p* **	**β (95% CI)**	** *p* **	**β (95% CI)**	** *p* **
MIS (continuous variable)	−0.32 (−0.49, −0.15)	<0.001	−0.52 (−0.78, −0.27)	<0.001	−0.43 (−0.70, −0.16)	0.002	−0.46 (−0.77, −0.14)	0.005
	**OR (95% CI)**	** *p* **	**OR (95% CI)**	** *p* **	**OR (95% CI)**	** *p* **	**OR (95% CI)**	** *p* **
MIS ≥ 6 vs. MIS < 6 (categorical variable)	0.47 (0.30, 0.75)	0.002	0.35 (0.18, 0.68)	0.002	0.33 (0.16, 0.69)	0.003	0.41 (0.19, 0.89)	0.025

* Handgrip strength (HGS) value is the maximum value of handgrip strength measured twice on each hand. The odds ratios (OR) and beta coefficients are standardized with 1 standard deviation increase in HGS. ^1^ Multivariable model 1 was adjusted for age, sex, history of diabetes mellitus and hypertension, stage of chronic kidney disease, smoking status, and overhydration. ^2^ Multivariable model 2 was adjusted for age, sex, history of diabetes mellitus and hypertension, stage of chronic kidney disease, smoking status, overhydration, income status, and education status. MIS, malnutrition–inflammation score; CI, confidence interval.

## Data Availability

The datasets generated and analyzed during this study are available from the corresponding author upon reasonable request. The data are not available publicly due to being part of an ongoing study.

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
