# Peer review of "Association of Handgrip Strength and Nutritional Status in Non-Dialysis-Dependent Chronic Kidney Disease Patients: Results from the KNOW-CKD Study"

_nutrients, 2024, doi:10.3390/nu16152442_

Round 1
Reviewer 1 Report
Comments and Suggestions for Authors
The manuscript under the title "Assessment of Nutritional Status in Non-Dialysis-Dependent Chronic Kidney Disease Patients Using Handgrip Strength: Results from the KNOW-CKD Study" is a very interestind study, well conducted and with valuable results.
There are some comments that need to be addressed.
1. Please reconsider the title as you are not assessing nutritional status with HGS but you correlate HGS with nutritional assessment results.
2. Do you have data of body composition to make the results of your study more robust?
3. Did you include patients receiving corticosteroids? If yes please exclude them from the final anaylsis as the effect of corticosteroids on muscle strength may be more profound than the effect of nutrition.
4. Do you have data regarding nutritional intake other than the results from MIS? If yes provide more information on current nutritional intake
5. You enrolled hospitalised patients in the study. Did you record the lenght of stay prior to the measurements? If yes please check if there is an effect on HSG, especially if surgeries or other interventions were done to them
Author Response
The manuscript under the title "Assessment of Nutritional Status in Non-Dialysis-Dependent Chronic Kidney Disease Patients Using Handgrip Strength: Results from the KNOW-CKD Study" is a very interested study, well conducted and with valuable results. There are some comments that need to be addressed.
Comment R1-1. Please reconsider the title as you are not assessing nutritional status with HGS but you correlate HGS with nutritional assessment results.
Response R1-1. Thank you for your comments. According to your comment, we revised the title of our study to: “Association of Handgrip Strength and Nutritional Status in Non-Dialysis-Dependent Chronic Kidney Disease Patients: Results from the KNOW-CKD Study".
Comment R1-2. Do you have data of body composition to make the results of your study more robust?
Response R1-2. Thank you for your comments. We have data on body composition assessed by the bioelectrical impedance analysis (BIA) in the KNOW-CKD phase II study. Based on your comment, we conducted a multivariate correlation analysis between the handgrip strength (HGS) and various factors including body mass index (BMI), skinfold thickness (SFT) measured from four areas, mid-arm muscle circumference (MAMC), serum albumin, hemoglobin, and data of BIA including intracellular water (ICW), extracellular water (ECW), total body water (TBW), fat mass, fat-free mass (FFM), and soft lean mass (SLM). This has been specified in the ‘Materials and Methods’ section of the revised manuscript, page 4, paragraph 2, line 162-163.
In the ‘Materials and Methods’ section:
2.5. Statistical analysis
“A multivariate correlation analysis was also conducted to evaluate the correlation between HGS and various nutritional parameters”
In multivariate correlation analysis, nutritional parameters including BMI, MAMC, ICW, ECW, TBW, FFM, SLM, and hemoglobin showed significantly positive correlation with HGS. Meanwhile, biceps and triceps SFT showed significantly negative correlation with HGS. This has been specified in the ‘Results’ section and Table S1 of the revised manuscript, page 7, paragraph 1, line 214-216.
In the ‘Results’ section:
3.2. Correlation between HGS and nutritional parameters
“In multivariate correlation analysis, HGS was positively correlated with BMI, MAMC, ICW, ECW, TBW, FFM, SLM, and hemoglobin. Meanwhile, HGS was negatively correlated with biceps and triceps SFT (Table S1).”
Furthermore, please kindly note that MIS already includes following components: 1) decreased fat stores assessed by physical examination, 2) muscle wasting assessed by physical examination, 3) BMI, 4) Serum total iron binding capacity (TIBC). Therefore, we carefully did not make additional adjustment with following nutritional parameters, although these were significantly correlated with HGS in a multivariate correlation analysis: 1) biceps and triceps SFT, which are used to assess body fat, 2) MAMC, FFM, and SLM, which are used to assess muscle mass, 3) BMI, 4) hemoglobin, which is closely associated with TIBC and can eliminate the effect of TIBC on MIS if adjusted. Finally, we aimed to focus on hydration status, which can be assessed using ICW, ECW, and TBW, as an additional adjustment variable.
From the perspective of hydration status, overhydration is a common complication in non-dialysis-dependent CKD patients [1] and is significantly associated with malnutrition status [2, 3]. Therefore, we aimed to additionally adjust for overhydration status when evaluating the association between HGS and MIS. We defined overhydration status as ‘ECW/TBW ratio >0.39’ [3] and ‘TBW/FFM ratio >0.74’ [4]. Finally, after including overhydration status as a covariate in multivariable model, we still demonstrated that higher HGS was associated with a lower risk of malnutrition.
[1] Guo Y, Zhang M, Ye T, Wang Z, Yao Y. Application of Bioelectrical Impedance Analysis in Nutritional Management of Patients with Chronic Kidney Disease. Nutrients. 2023 Sep 12;15(18):3941.
[2] Wang WL, Liang S, Zhu FL, Liu JQ, Chen XM, Cai GY. Association of the malnutrition-inflammation score with anthropometry and body composition measurements in patients with chronic kidney disease. Ann Palliat Med. 2019 Nov;8(5):596-603.
[3] Eyre S, Stenberg J, Wallengren O, Keane D, Avesani CM, Bosaeus I, Clyne N, Heimbürger O, Indurain A, Johansson AC, Lindholm B, Seoane F, Trondsen M; SWEBIS network. Bioimpedance analysis in patients with chronic kidney disease. J Ren Care. 2023 Sep;49(3):147-157.
[4] Wang Z, Deurenberg P, Wang W, Pietrobelli A, Baumgartner RN, Heymsfield SB. Hydration of fat-free body mass: review and critique of a classic body-composition constant. Am J Clin Nutr. 1999 May;69(5):833-41.
Comment R1-3. Did you include patients receiving corticosteroids? If yes please exclude them from the final analysis as the effect of corticosteroids on muscle strength may be more profound than the effect of nutrition.
Response R1-3. Thank you for your comments. We apologize for omitting the information that patients who had taken immunosuppressants within the past year were excluded from our study. Therefore, patients receiving corticosteroids were also excluded from our study. This has been specified in the ‘Materials and Methods’ section of the revised manuscript, page 2, paragraph 4, line 84-88.
In the ‘Materials and Methods’ section:
2.1. Study setting and population
“Patients with a diagnosis of glomerulonephritis or polycystic kidney disease, a previous history of dialysis, any organ transplant, heart failure (New York Heart Association functional classification III or IV), liver cirrhosis (Child-Pugh class B or C), or cancer, pregnant women, and those who had taken immunosuppressants within the past year were excluded from this study.”
Comment R1-4. Do you have data regarding nutritional intake other than the results from MIS? If yes provide more information on current nutritional intake
Response R1-4. Thank you for your comments. We have data on nutritional intake assessed by a food frequency questionnaire in the KNOW-CKD phase II study. Based on your comment, we compared total energy, protein, fat, and carbohydrate intake between the low and high HGS groups. Additionally, we compared protein, fat, and carbohydrate intake as a percentage of total energy. There were no significant differences in nutritional intake between the low and high HGS groups. These results are presented in the ‘Results’ section and Table 1 of the revised manuscript, page 5, paragraph 1, line 196-198.
In the ‘Results’ section:
3.1. Baseline characteristics
“There were no significant differences between the low and high HGS groups in total energy intake and macronutrients intake including protein, fat, and carbohydrate.”
Comment R1-5. You enrolled hospitalized patients in the study. Did you record the length of stay prior to the measurements? If yes please check if there is an effect on HSG, especially if surgeries or other interventions were done to them
Response R1-5. Thank you for your comments. Please note that we enrolled patients who visited the outpatient department. Therefore, we do not have data on the length of hospital stays.

Reviewer 2 Report
Comments and Suggestions for Authors
Handgrip strength (HGS) is suggested as an indirect assessment of nutritional status in chronic kidney disease (CKD) patients, but evidence is limited for non-dialysis-dependent CKD (NDD-CKD) patients. This cross-sectional study included patients from the Phase II Korean Cohort Study for Outcome in Patients With CKD. Asa a result, authors suggested that HGS showed fair predictability for malnutrition in men and women. In addition, HGS is a useful diagnostic indicator of malnutrition in NDD-CKD patients.
This manuscript is written well; however, this will be able to revise and reconsider several points.
1) Please explain in more detail in the introduction the significance of measuring grip strength, including its relationship to other diseases.
2) Regarding the significance of measuring grip strength, please add a note about other diseases and older subjects in chronic phase, such as heart disease, in addition to diabetes and CKD, from the perspective of upper limb function.
Longitudinal Changes of Handgrip, Knee Extensor Muscle Strength, and the Disability of the Arm, Shoulder and Hand Score in Cardiac Patients During Phase II Cardiac Rehabilitation - PubMed (nih.gov)
Changes in Physical and Psychological States with Respect to the Gender of Outpatients Receiving Rehabilitation at Geriatric Health Services Facilities during the COVID-19 State of Emergency - PubMed (nih.gov)
3) Please show the strengths and weaknesses of this study.
Comments on the Quality of English Language
Minor editing of English language required
Author Response
Comments and Suggestions for Authors
Handgrip strength (HGS) is suggested as an indirect assessment of nutritional status in chronic kidney disease (CKD) patients, but evidence is limited for non-dialysis-dependent CKD (NDD-CKD) patients. This cross-sectional study included patients from the Phase II Korean Cohort Study for Outcome in Patients With CKD. As a result, authors suggested that HGS showed fair predictability for malnutrition in men and women. In addition, HGS is a useful diagnostic indicator of malnutrition in NDD-CKD patients. This manuscript is written well; however, this will be able to revise and reconsider several points.
Comment R2-1. Please explain in more detail in the introduction the significance of measuring grip strength, including its relationship to other diseases
Response R2-1. Thank you for your comments. Following your suggestions, we explained in more detail on the association between handgrip strength and other diseases, such as diabetes, hypertension, metabolic syndrome, cardiovascular disease, and stroke, in the introduction section. Furthermore, we included more information on the relationship of handgrip strength with mortality and adverse renal outcomes, particularly in chronic kidney disease (CKD) patients. These revisions have been specified in the ‘Introduction’ section of the revised manuscript, page 2, paragraph 1, line 46-62.
In the ‘Introduction’ section:
“Among various nutritional assessment tools, handgrip strength (HGS) is a simple and reliable method for evaluating muscle function and overall physical fitness. In previous studies, handgrip strength (HGS) has been linked to a number of different diseases. A low HGS is often indicative of diminished muscle quality and reduced muscle mass, which can result in metabolic dysregulation and insulin resistance. In previous studies, low HGS has been shown to be associated with higher risks of type 2 diabetes mellitus (DM), hypertension, and metabolic syndrome. Given that low HGS is associated with these metabolic diseases, previous studies have also shown that low HGS is associated with an increased risk of cardiovascular diseases and stroke. Moreover, HGS has been used to evaluate physical status and activities associated with the upper limb in specific populations, including elderly participants and those with cardiac diseases.
Particularly, in CKD patients, low HGS was associated with a higher risk of mortality and renal outcomes in previous studies. In light of the established link between HGS and adverse outcomes in patients with CKD, HGS has been employed in a range of clinical settings, including the assessment of nutritional status in this patient population.”
Comment R2-2. Regarding the significance of measuring grip strength, please add a note about other diseases and older subjects in chronic phase, such as heart disease, in addition to diabetes and CKD, from the perspective of upper limb function.
Longitudinal Changes of Handgrip, Knee Extensor Muscle Strength, and the Disability of the Arm, Shoulder and Hand Score in Cardiac Patients During Phase II Cardiac Rehabilitation - PubMed (nih.gov)
Changes in Physical and Psychological States with Respect to the Gender of Outpatients Receiving Rehabilitation at Geriatric Health Services Facilities during the COVID-19 State of Emergency - PubMed (nih.gov)
Response R2-2. Thank you for your comments and thank you for introducing meaningful studies related to handgrip strength (HGS). Following your suggestions, we have added content about the evaluation of physical status using HGS in elderly participants and those with heart diseases, from the perspective of upper limb function. This has been specified in the ‘Introduction’ section of the revised manuscript, page 2, paragraph 1, line 54-57.
In the ‘Introduction’ section:
“Moreover, HGS has been used to evaluate physical status and activities associated with the upper limb in specific populations, including elderly participants and those with cardiac diseases.”
Comment R2-3. Please show the strengths and weaknesses of this study.
Response R2-3. Thank you for your comments and thank you for giving us the opportunity to summarize the strengths and weaknesses of our study once more in the response letter. Based on your comment, we summarized the strengths and weaknesses of our study in one paragraph in the ‘Discussion’ section of the revised manuscript, page 11, paragraph 2, line 358-373.
In the ‘Discussion’ section:
“Our study has the strength of investigating the association between HGS and malnutrition status, defined by MIS, in the largest number of NDD-CKD patients to date. Furthermore, our study is the first to propose cut-off values for HGS to predict malnutrition status in NDD-CKD patients. These strengths provide significant evidence for assessing nutritional status using HGS, especially in a context where methods for evaluating nutritional status in NDD-CKD patients are limited and existing evidence is scarce. This study has several limitations. First, due to the observational and cross-sectional nature of the study, a causal relationship between HGS and malnutrition cannot be established and residual confounding factors and reverse causation may be present. Second, the study was conducted using data from a single country; therefore, further studies with diverse ethnic groups are necessary to expand the generalizability of the findings. Third, as mentioned in the methods section, a definitive MIS threshold for the diagnosis of malnutrition has not yet been established. Although we applied different MIS thresholds to assess malnutrition in our sensitivity analysis and obtained similar results, further research is necessary to investigate the criteria of MIS associated with adverse outcomes in NDD-CKD patients through longitudinal analysis.”

Round 2
Reviewer 1 Report
Comments and Suggestions for Authors
Thank you for addressing all my comments sufficienlty